# Soil Structure and Ectomycorrhizal Root Colonization of Pecan Orchards in Northern Mexico

**DOI:** 10.3390/jof9040440

**Published:** 2023-04-04

**Authors:** Hilda Karina Sáenz-Hidalgo, Juan Luis Jacobo-Cuellar, Erick Zúñiga-Rodríguez, Graciela Dolores Avila-Quezada, Víctor Olalde-Portugal, Abeer Hashem, Elsayed Fathi Abd_Allah

**Affiliations:** 1Centro de Investigación en Alimentación y Desarrollo, Chihuahua 33089, Mexico; 2Facultad de Ciencias Agrotecnológica, Universidad Autónoma de Chihuahua, Campus 1, Chihuahua 31000, Mexico; 3Cinvestav-Instituto Politécnico Nacional, Guanajuato 36824, Mexico; 4Botany and Microbiology Department, College of Science, King Saud University, P.O. Box. 2460, Riyadh 11451, Saudi Arabia; 5Plant Production Department, College of Food and Agricultural Sciences, King Saud University, P.O. Box. 2460, Riyadh 11451, Saudi Arabia

**Keywords:** *Carya illinoinensis*, phosphorus, ectomycorrhizae, pecan tree, *Pisolithus arenarius*, *Pisolithus tinctorius*

## Abstract

Pecan trees form a symbiotic relationship with ectomycorrhizal fungi (ECM), which actively provide nutrition to the roots and protect them from phytopathogens. Although these trees originated in the southern United States and northern Mexico, information on their root colonization by ECM is insufficient in terms of a representative number of samples, both in these regions and worldwide. Therefore, the objectives of this study were to determine the percentage of ectomycorrhizal colonization (ECM) of pecan trees of different ages in conventional and organic agronomic orchards and to identify ectomycorrhizal sporocarps, both morphologically and molecularly. The rhizospheric soil properties and the ECM percentages were analyzed for 14 Western variety pecan tree orchards between 3 and 48 years of age and grouped according to the agronomic management method. DNA extraction, internal transcribed spacer amplification, and sequencing were conducted on the fungal macroforms. The ECM colonization percentage fluctuated between 31.44 and 59.89%. Soils with low phosphorus content showed higher ECM colonization. The ECM concentrations were relatively homogeneous in relation to the ages of the trees, and organic matter content did not affect the percentage of ECM colonization. The highest ECM percentages occurred with the sandy clay crumb texture soil, with an average of 55% ECM, followed by sandy clay loam soils with 49.5%. The *Pisolithus arenarius* and *Pisolithus tinctorius* fungi were molecularly identified from sporocarps associated with pecan trees. This is the first study that reports *Pisolithus arenarius* as being associated with this tree.

## 1. Introduction

Pecan trees (*Carya illinoinensis* (Wangeh) K. Koch) are native to North America, although they are cultivated globally for their nutritional and economic value [1,2,3]. They are one of the most economically viable crops in both northern Mexico and the southern United States. Mexico was the second-largest producer of pecan nuts worldwide in 2017, with a production of 147,198 tons [4], while the United States produced 136,000 tons [5].

The Mexican state of Chihuahua is the largest producer in the country and contributes 57% of the national production [6]. Chihuahua’s pecan production was 92,938 tons in 2017 [4], which was the greatest amount of any state globally, followed by the state of Georgia in the USA with 48,534 tons in that year [5].

The soil ecosystem plays an important role in agricultural production. In soils where rhizosphere microorganisms such as native bacteria [7] and fungi, including mycorrhizal fungi [8], are abundant [9] and the availability and absorption of mineral nutrients are enhanced [10,11]. Ectomycorrhizal fungi (ECM) absorb and transport dissolved nutrients to plants in a symbiotic manner, whereby the plants have access to sources of nitrogen, organic phosphorus, and other essential nutrients mobilized by the ECM [12]. ECM depend on plant photoassimilates and manipulate root morphology through signaling molecules. ECM manipulates plant hormone receptors to inhibit the defenses of the plant and facilitate fungal colonization [13]. The symbiotic association between specialized hyphae and plant roots [14] confers some resistance to the plant against root pathogens [15,16]. Furthermore, the hyphae extend outward from the roots and are able to absorb additional nutrients, mainly phosphorus [11]. Ectomycorrhizae modify and surround the roots to form a mycelium mantle [17].

ECM and the plant family Juglandaceae, including the genus *Carya,* often form symbiotic associations [18]. In addition, the heat-tolerant and drought-tolerant genus *Pisolithus* forms a symbiosis with pecan tree roots [19,20,21,22,23].

Several factors are involved in the symbiotic association between ECM and pecan tree roots. For example, the percentage of colonization and the ECM structures in pecan roots in Chihuahua, Mexico, can vary according to the climatic and edaphic conditions in the orchard [24]. In addition, edaphic factors such as pH, phosphorus content, organic fertilizers, and plant cover can exert important effects on root colonization [24,25,26].

Moreover, the age of a tree is an important factor since the production or biomass of mycorrhizae increases as the tree matures [27]. Studies on the effect of organic and conventional orchard management on the diversity of microorganisms in the soil have demonstrated that organic management promotes an increase in the diversity of arbuscular mycorrhizal fungi [28,29]. The general assumption is that a diversity of microorganisms exists, including ECM, according to the agronomic management of pecan orchards; however, there is currently limited data on the subject [21,30], some of which focused on other tree species [31,32]. There are no identified studies on the molecular identification of sporocarps inhabiting pecan orchards in Mexico or the USA.

Studies in Europe, where pecan trees are alien, have shown that hickory trees can readily establish symbioses with local ECM fungi [33,34,35]. Therefore, these symbioses are likely to also occur between trees and the ECM of their native regions. Mycorrhizal colonization of pecan trees is clearly beneficial for tree production, and several factors can influence this symbiosis. Therefore, the objectives of this study were (a) to determine the ectomycorrhizal colonization percentage of absorbing roots of pecan trees of various ages in both conventional and organic agronomic management orchards and (b) to morphologically and molecularly identify ectomycorrhizal sporocarps. The information obtained can be used to design future research strategies and strengthen integrated crop management programs.

## 2. Materials and Methods

### 2.1. Experimental Site

The study included 14 pecan orchards in Chihuahua State, Mexico (Table 1). Samples of absorbing roots were collected from March to August 2020 to analyze ectomicorrhization. Western Schely variety pecan tree orchards on Creole rootstock were grouped according to type of agronomic management (organic or conventional) and tree age. The orchards with trees from 3 to 20 years old that were planted at 8 × 8 m in a real frame were grouped as “young orchards” (Figure 1). The orchards with trees from 21 to 48 years old and planted at 13 × 13 m in a staggered pattern were grouped into “adult orchards”. Homogeneity was observed in the soil characteristics within and among orchards with slopes ≤2%.

In the Carmen MA, Carmen SG, and San Jorge orchards, a commercial product was previously applied that contained *Pisolithus tinctorius* (1 × 10^6^ spores g^−1^), *Glomus intraradices* (1 × 10^3^ spores g^−^^1^)*, Azospirillum brasilense* (1 × 10^6^ CFU g^−^^1^), and 20% total oxidizable organic carbon.

### 2.2. Rhizospheric Soil Analysis

Rhizospheric soil samples were obtained from the 4 cardinal points of three trees at a depth of 30 cm, following the methodology of Cruz-Álvarez et al. [36]. According to Sanchez et al. [24], pecan tree ECM roots are located at a depth of 5–35 cm.

A total of 4 100-g subsamples were collected per tree and combined to prepare a mixed sample for each orchard. Soil samples were dried at room temperature and sieved. Texture analysis was performed using the hydrometer technique with 50 g of rhizospheric soil [37]. Phosphorus content (P) was determined using 1 g of rhizospheric soil with 20 mL of 0.5 molar sodium bicarbonate (NaHCO_3_) at a pH of 8.5. The mixture was shaken for 30 min, filtered, and measured at 880 nm. The results are expressed in ppm [38]. Organic matter was quantified by combining 0.30 g of rhizospheric soil with 10 mL of a 0.17 M potassium dichromate solution and 10 mL of concentrated sulfuric acid and shaking for 1 min. The solution was cooled to a still volume of 100 mL and 5 mL, of concentrated phosphoric acid was added. After 10 min, 2 to 3 drops of diphenylamine indicator were added, the solution was titrated with 1 M ferrous sulfate [39], and soil pH was tested with a glass electrode in a soil–water ratio of 1:2.5 (*w*/*w*).

### 2.3. Mycorrhizal Colonization of Pecan Trees

Three pecan trees were randomly selected per orchard, with a minimum distance of 40 m between trees. The selected trees were uniform in size and trunk diameter. Three absorbing roots of 10–15 cm each were collected at a soil depth of 5–35 cm at the four cardinal points of the tree root [36].

The roots were washed and observed under a stereomicroscope to record the type of ECM branching, which was classified according to morphological structure based on the criteria established by Marx et al. [40]. ECM colonization was determined from 3 10-cm-long root portions from each tree. Each 10 cm represented 1 of the 3 repetitions [41]. Nine means were obtained per orchard.

### 2.4. Sporocarp Collection—Molecular Genetic Identification

In August 2020, fungal macroforms were observed only in the La Concha orchard. Internal samples were obtained in the laboratory from two fruiting bodies of different colors, and DNA extraction was performed using a commercial kit (Ultraclean Isolation DNA Kit, MoBio brand, San Mateo, CA, USA) following the manufacturer’s instructions. The quality and concentration of the DNA were verified using a Nanodrop 2000c spectrophotometer. The internal transcribed spacer (ITS) gene region, 18S rDNA (partial sequence) gene, 5.8S rDNA gene, internal transcribed spacers 1 and 2 (full sequence), and the 28S rDNA gene (partial sequence) were amplified by PCR using the ITS5 (5′-GGAAGTAAAAGTCGTAACAAGG-3′) [42] and ITS4 5′-TCCTCCGCTTATTGATATGC-3′) primers [42].

The protocol included an initial denaturation at 94 °C for 1 min, followed by 40 cycles each of 94 °C (1 min), 50 °C (2 min), and 72 °C (1 min), and a final extension at 72 °C (5 min). PCR was conducted using a Peltier Thermal Cycler PTC-200 (Bio-Rad, Hercules, CA, USA), and the PCR products were verified by electrophoresis on a 1.2% agarose gel. PCR products were sequenced in both directions using an Applied Biosystems Model 3730XL Automated DNA Sequencing System.

### 2.5. Phylogenetic Analysis of the ITS Region

Consensus sequences for each macroform were analyzed against reference sequences using the NCBI BLAST bioinformatics tool. All sequences, including GenBank references, were compiled into a single file (in fasta format) which was aligned using the Clustal W algorithm [43] within the MEGA7 software [44].

Subsequently, phylogenetic reconstruction was performed for the rDNA ITS dataset using the maximum likelihood method and the 2-parameter model of Kimura [45]. The tree with the highest log-likelihood (−2057.29) is shown. The percentage of trees in which associated taxa clustered is shown below the branches. The initial trees for the heuristic search were obtained automatically by applying the Neighbor-Join and BioNJ algorithms to a matrix of pairwise distances estimated using the composite maximum likelihood (MCL) approach and selecting the topology with the highest likelihood. This analysis involved 13 nucleotide sequences. The codon positions included were 1st + 2nd + 3rd + No. coding. There were a total of 594 positions in the final data set. Evolutionary analyses were performed using MEGA11 [46].

The two sequences retrieved were compared with those of *Pisolithus tinctorius, Pisolithus arenarius*, and *Pisolithus arrhizus,* which are located in the GenBank database at NCBI. *Scleroderma citrinum* was designated as the outgroup for the construction of the evolutionary tree.

### 2.6. Statistical Analysis

The percentage of ECM colonization was analyzed using the non-parametric Mann–Whitney test with correction for ties and 95% confidence. Correlations were analyzed among the percentage of ectomycorrhization, age of the orchard, type of agronomic management, concentration of P, and level of organic matter using the Minitab program version 19.2020.1.0 (© 2023 Minitab).

## 3. Results

### 3.1. Percentage of Ectomycorrhizal Colonization

The percentage of root tips colonized by ECM in pecan trees was from 31.4 to 59.9% in the 14 orchards (Figure 2). Virtually all portions of roots observed in the stereoscope had varying degrees of ectomycorrhizal colonization.

### 3.2. Ectomycorrhization and Agronomic Management-Tree Age

Regarding the agronomic management of trees older than 21 years, significant statistical differences were found in the ectomycorrhization percentage (*p* ≤ 0.05). The highest colonization (52.78%) (*n* = 4) in adult trees occurred in orchards with organic management and the lowest (39.04%) (*n* = 3) in those with conventional management (Figure 2).

In orchards under 20 years old, there was no statistically significant difference in the percentage of colonization (*p* > 0.05) between orchards with conventional (48.61%) and organic (46.26%) agronomic management methods.

The application of commercial ECM in the Carmen MA, Carmen SG, and San Jorge orchards did not significantly affect the colonization percentage; thus, it can be deduced that native ectomycorrhizal fungi are competitive.

### 3.3. Ectomycorrhization and Soil Properties

#### 3.3.1. Phosphorus (P)

The P concentration had a significant effect on the ectomycorrhization of the rhizospheric soil (*p* ≤ 0.05). In adult orchards, a higher ECM colonization was observed in tree roots with low P contents in the rhizosphere ≤ 22 ppm (Figure 3). However, in young orchards, there was no significant effect, although changes could eventually be reflected after consistently applying agronomic management.

#### 3.3.2. Influence of Organic Matter (OM) and Phosphorus (P) on ECM

The orchards were grouped according to the OM and P content of the soil, resulting in three groups: OM < 1 low P; OM > 1 low P; and OM > 1 high P. In this study, the OM content did not influence ECM colonization. In contrast, the P content affected ectomycorrhization (*p* ≤ 0.05), with a greater colonization by ECM in orchards with P ≤ 22 ppm (Figure 4).

#### 3.3.3. Texture

The highest ECM percentages correlated with sandy clay crumb texture soils with an average of 55% ECM, followed by clay loam soils with 49.5% ECM. The soils with loam and clayey texture had the lowest levels of ECM, with 31.4 and 34%, respectively (Table 2).

#### 3.3.4. pH

There was no correlation between the ECM percentage and pH of the rhizospheric soil of the pecan trees. The pH in the calcareous soils of the pecan orchards in this study ranged from 7.58 to 8.02.

### 3.4. Ectomycorrhizal Morphological Structure

In this study, a diversity of roots that were structurally modified by ectomycorrhization was observed. They consisted of simple, coralloid, monopodial pyramidal, and dichotomous roots (Figure 5). The most common was the simple structure.

### 3.5. Sporocarps

The identified macroforms belonged to La Concha orchard samples. Their sporocarps were globular and 11 cm high, the roots were 6 cm long, the fruiting body gradually changed color (brown-black), and the peridioles near the top were enlarged and opened to expose a powdery colored gleba (Figure 6b). These characteristics coincided with those of *Pisolithus tinctorius* as described by Razzaq and Shahzad [47]. The features of the second sporocarp (Figure 6a) were similar.

### 3.6. Molecular-Genetic Identification

The amplified region of the ITS genes showed a 99.2% similarity with *Pisolithus arenarius.* The second sequence was identified as *Pisolithus tinctorius* (99.2% similarity) using the NCBI BLAST tool for sequence similarity searches.

The phylogenetic tree, based on 647 bp fragments of the ITS gene, was developed using the maximum likelihood evolutionary analysis method (Figure 7).

The evolutionary history was inferred using the maximum likelihood method and the Kimura 2-parameter model [45], which produced the tree with the highest log probability (−2057.29). The percentage of trees in which associated taxa clustered is shown below the branches. The initial trees for the heuristic search were obtained automatically by applying the Neighbor-Join and BioNJ algorithms to a matrix of pairwise distances estimated using the composite maximum likelihood (MCL) approach and then selecting the topology with the higher likelihood value.

## 4. Discussion

The study showed for the first time that ectomycorrhized roots are present in a high number of pecan trees: 42 individual trees in 14 orchards.

### 4.1. Percentage of Ectomycorrhizal Colonization

Previous studies of pecan trees in Chihuahua, Mexico, have shown data similar to our results in relation to the percentage of ECM, although only the results of four previously studied orchards are reported. The study by Sánchez et al. [24] estimated a 6 to 46% ECM colonization in the roots of adult pecan trees in two orchards in the Delicias and Rosales counties of Chihuahua.

In addition, Olivas-Tarango et al. [21] reported percentages of ectomycorrhization by *Pisolithus* sp. of 40 to 70% in a western pecan orchard in Conchos County, Chihuahua, Mexico. Those authors attributed the high percentages of colonization to the application of edaphic zinc. Tarango-Rivero et al. [30] found 67 to 83% of ECM (*Pisolithus* sp.) in a pecan orchard in Delicias, Chihuahua, under various treatments.

Our study provides more robust results than those previously conducted in northern Mexico and the southern USA since ectomycorrhization was analyzed in pecan tree roots of different ages using different agronomic management methods and a greater number of orchards (14).

Although a diversity of spores was found in the rhizospheric soil analyzed in our study, including those of the genus *Glomus*, no endomycorrhizal fungi were located. However, a previous study by Muñoz-Márquez et al. [20] reported the genus *Glomus* endomycorrhizing pecan tree roots. Competition between arbuscular mycorrhizae (e.g., *Glomus mosseae*) and ECM fungi (e.g., *Hebeloma leucosarx*) has been documented, along with the simultaneous colonization by both species on root tips of deciduous trees such as *Salix repens* [48]. That study reported a rate of up to 5% colonization by arbuscular mycorrhizae compared to 70% colonization by ectomycorrhizae.

Rudawska et al. [33] found dual mycorrhizal colonization by endo (AM) and ECM fungi on the roots of *Carya* spp. ECM colonization was between 11.3 and 16.9%, while AM colonization was 8.6–12.2%. Those authors observed that AM were the first to establish symbiosis with *Carya*; however, these were gradually replaced by ECM as the trees aged.

During simultaneous colonization, the ectomycorrhizae may dominate the internal area of the roots over the endomycorrhizae. For future studies, a greater number of samples should be analyzed to detect colonization by *Glomus* sp. and other AM in pecan trees, along with a metagenomic analysis of colonized roots to obtain a broad understanding of the populations that interact with the roots of these trees.

### 4.2. Ectomycorrhization and Agronomic Management-Tree Age

According to Wallander and Ekblad [27], as the tree matures, mycorrhizal production increases, which corroborates the results obtained with the adult organic orchards in our study.

### 4.3. Ectomycorrhization and Soil Properties

#### 4.3.1. Phosphorus (P) and ECM

An ectomycorrhizal fungus, when faced with low phosphorus conditions, activates enzymes to induce symbiosis with a tree, since low amounts of nutrients can serve as signals [49,50].

Thus, in soils with high concentrations of P, a decrease in the abundance of fungi has been reported, suggesting that plants may depend less on ECM in soils with high P availability [25,51]. The interaction begins with the production of signal phytases that detect the ectomycorrhizal fungus, and free phosphate is acquired by means of a phosphate transporter (gene subfamily Pht1) of the ECM plasmatic membrane [52].

#### 4.3.2. Influence of Organic Matter (OM) on ECM

In our study, the OM content did not influence ECM colonization, whereas the P content affected ectomycorrhization. Lower phosphorus contents resulted in greater ECM colonization. These results demonstrate that P content plays an important role in ECM establishment in pecan trees.

#### 4.3.3. Influence of Texture in ECM

The highest ECM percentages were correlated with the sandy clay crumb texture soil. Moreover, the El Maguey orchard consisted of compacted soil and stone. This may be related to a lower percentage of ectomycorrhizae because the porosity of the soil affects the elongation of the ectomycorrhizae and the growth of the mycelium by limiting their vital space [53,54].

#### 4.3.4. pH

The soil pH of the orchards in our study ranged from 7.58 to 8.02, irrespective of ECM percentage. The pH of the calcareous soils where the pecan tree was established was high in all the orchards in this study. Olivas-Tarango et al. [21] reported a pH of 7.5 in pecan tree soils in Chihuahua and a high presence of ectomycorrhized roots. In the study by Becerra et al. [48], the greatest ectomycorrhizal colonization of *Alnus acuminata* occurred in soil with a pH of 6.6; in this study, the ECM percentage was measured in soils with pHs ranging from 5.6 to 6.6.

According to the results of our study, ectomycorrhizae support alkaline soils and may perform a symbiotic function with pecan trees in relation to the high salt content. This finding provides useful information for future studies of ectomycorrhization in pecan trees under salt stress.

### 4.4. Ectomycorrhizal Morphological Structure

The root structures found in our study—simple, coralloid, monopodial pyramidal, and dichotomous—have been described by Agerer and Rambold [55]. In addition, Bonito et al. [18], Muñoz-Márquez et al. [41], and Sánchez et al. [24] found these ectomycorrhizal structures in the apical roots of pecan trees.

### 4.5. Sporocarps

Within the same species, the color and shape of the sporocarp, the color of the basal rhizomorphs, and the length of the stipe can vary [55]. The identified macroforms coincided with the typical characteristics of *Pisolithus tinctorius* described by Razzaq and Shahzad [47]. The characteristics of the second sporocarp were similar, although its colors were darker, perhaps due to its maturity.

*Pisolithus tinctorius* is a fungus commonly found ectomycorrhizing the roots of tree species such as eucalyptus [56], several pine species, Douglas fir, western hemlock, bur oak [40], and *Castanea sativa* [57].

Several studies on pecan trees have reported a wide diversity of ectomycorrhizal fungi of the genera *Tuber* and *Scleroderma* in ectomycorrhizal root tips that have been identified by sequencing of the ITS and LSU rDNA genes [23]. *Tomentella*, *Thelephora*, *Russula*, *Boletus*, and *Pisolithus* have also been identified [26,41].

A comparison of ITS sequences for *Pisolithus* isolates in the *Carya* GenBank nucleotide database (NCBI) showed that the genetically closest specimens were the three *Carya* specimens from the state of Chihuahua: *Pisolithus tinctorius* (Accession: OM780028), *Pisolithus arenarius* (Accession: OM780027), and *Pisolithus* sp. (Accession: FJ652047); the most genetically distant was the specimen from Brazil: uncultured *Pisolithus* (Accession: MT586545).

To the best of our knowledge, this is the first study reporting the ectomycorrhizal fungus *Pisolithus arenarius* in pecan orchards. This species had not previously been reported in association with the genus *Carya*. The two sequences of this study were deposited in GenBank (NCBI) with accession numbers OM780027 for *Pisolithus arenarius* and OM780028 for *Pisolithus tinctorius*. According to the MycoBank database (www.MycoBank.org), subspecies or forms have been reported; therefore, future studies should be performed to amplify the number of genes or the complete genome of each species found in this study.

### 4.6. Highlights

This study reports *Pisolithus arenarius* as a new ectomycorrhizal species associated with pecan trees.

The study is more robust than those previously conducted in northern Mexico and the southern United States because a greater number of pecan trees were analyzed.

We demonstrated that all roots of pecan trees were ectomycorrhized; the root structures were dichotomous branching, monopodial pyramidal, simple, coralloid, or simple and covered with a fungal blanket.

Soils poor in phosphorus (less than 22 ppm) promoted high ECM colonization. In adult trees, the greatest ectomycorrhization occurred in orchards with organic management. The organic matter content did not affect the percentage of ECM colonization.

## 5. Conclusions

For the first time, the percentage of ectomycorrhized *Carya* roots was determined using a large number of pecan trees.

The ectomycorrhizal pecan root tips were located at a depth of 5–35 cm, which is precisely where the nutrients are found. The results obtained allowed us to conclude that the pecan tree was colonized by ECM fungi, with colonization percentages of 31.4–59.9%. The variability of the ECM percentage can be influenced by several factors, particularly the phosphorus content, which was negatively related to ECM colonization. Similarly, adult trees with organic management had a positive correlation with the ECM percentage. This trend was not observed with young trees. The molecular techniques allowed for the discrimination among the species of these fungi (sporocarps), thereby demonstrating the presence of the heat- and drought-tolerant ECM *Pisolithus arenarius* and *Pisolithus tinctorius* in a pecan orchard. In this study, we report for the first time the association of *P. arenarius* with pecan trees.

## Figures and Tables

**Figure 1 jof-09-00440-f001:**
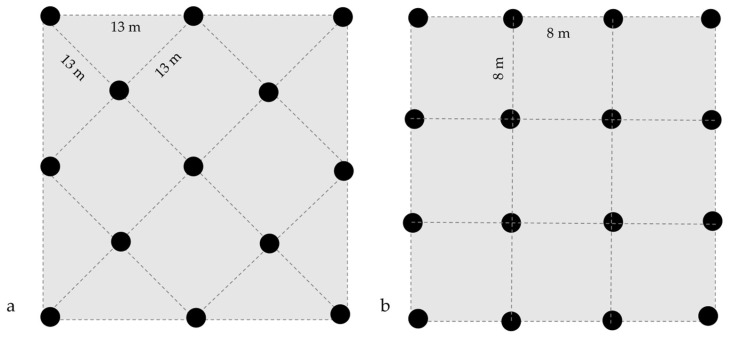
Pecan orchard planting frame. (**a**) Trees in staggered pattern and (**b**) trees in real frame.

**Figure 2 jof-09-00440-f002:**
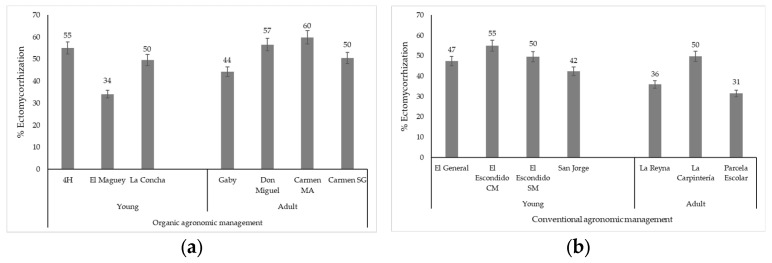
Correlation of agronomic management method and tree age with respect to the percentage of ECM (*p* ≤ 0.05) in the roots of Western Schley pecan trees [*Carya illinoinensis* (Wangeh) K. Koch] of 3 to 48 years old, established in Chihuahua State, Mexico. (**a**) Organic agronomic management; (**b**) conventional agronomic management. (*p* ≤ 0.05). Young = 3 to 20 years old. Adult = greater than 20 years old.

**Figure 3 jof-09-00440-f003:**
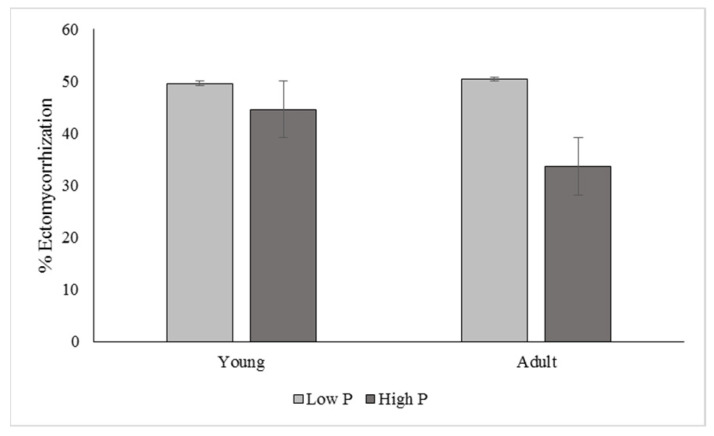
Tree age, ECM percentage (*p* ≤ 0.05), and phosphorus concentration (ppm) in rhizospheric soil of pecan trees established in Chihuahua State, Mexico. Young trees (<20 years), adult trees (>20 years).

**Figure 4 jof-09-00440-f004:**
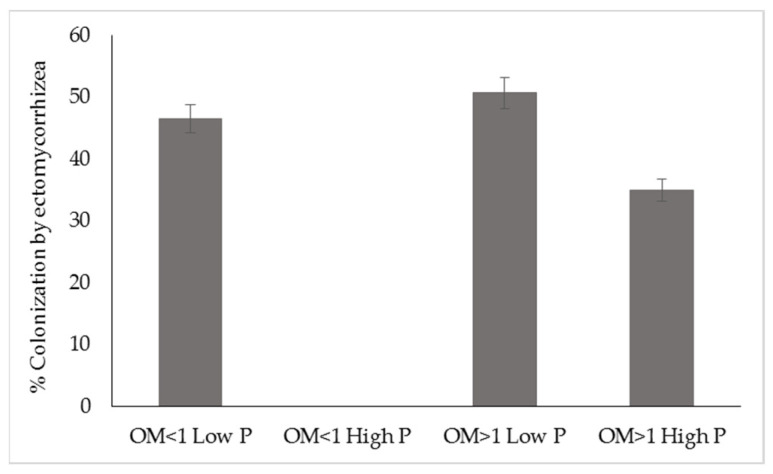
Correlation of the contents of organic matter (%) and phosphorus (ppm) in the rhizospheric soil with the percentage of ECM colonization (*p* ≤ 0.05) in the roots of pecan trees in orchards established in Chihuahua State, Mexico. OM < 1Low P = organic matter ≤ 1% and phosphorus ≤ 22 ppm. OM < 1 High P = organic matter ≤ 1% and phosphorus > 22 ppm. OM >1 Low P = organic matter > 1% and phosphorus ≤ 22 ppm. OM>1 High P = organic matter > 1% and phosphorus > 22 ppm.

**Figure 5 jof-09-00440-f005:**
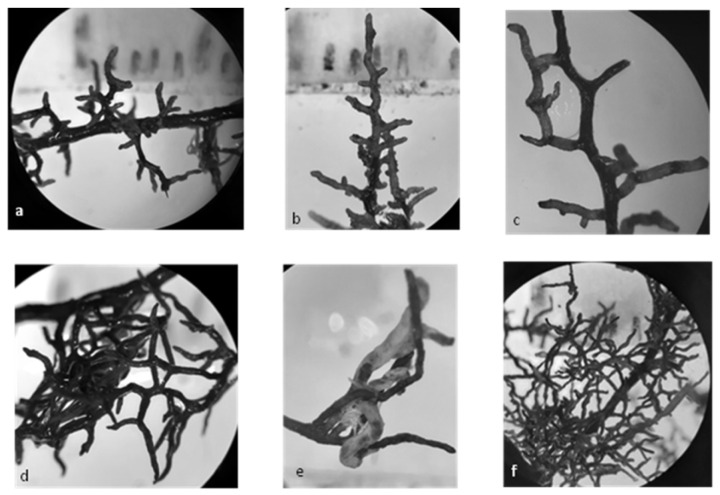
ECM branching types of *C. illinoinensis* (Wangeh) K. Koch in Chihuahua, Mexico. (**a**) Juvenile stage of dichotomous ramifications; (**b**) pyramidal monopodial; (**c**) simple; (**d**,**f**) coralloid; (**e**) simple, covered with a fungal mantle.

**Figure 6 jof-09-00440-f006:**
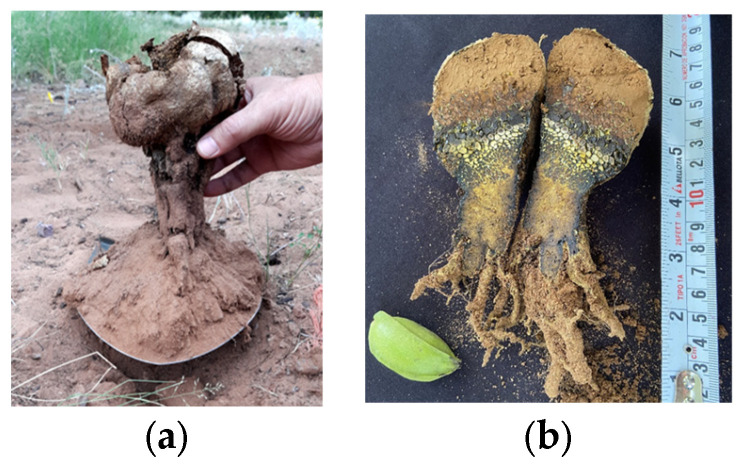
Fruiting bodies formed in the La Concha pecan orchard. (**a**) *Pisolithus arenarius* accession number OM780027 (2020) (**b**) *Pisolithus tinctorius* accession number OM780028 (2021); both are from the NCBI.

**Figure 7 jof-09-00440-f007:**
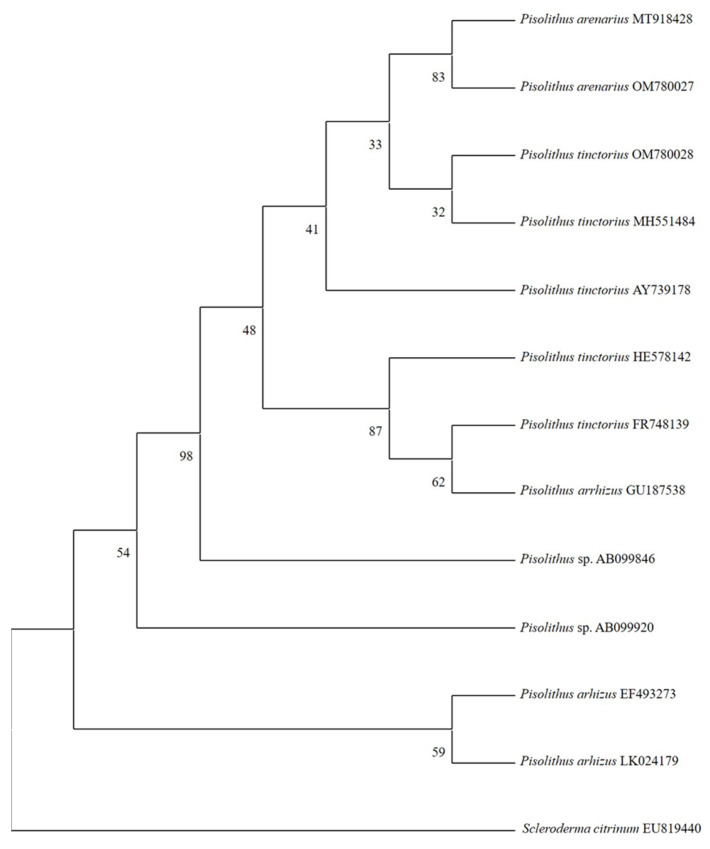
Phylogenetic tree of sequences constructed from DNA (ITS) to estimate relationships between species of *Pisolithus*. *Pisolithus arenarius* accession number OM780027 and *Pisolithus tinctorius* accession number OM780028 were the ECM of this study.

**Table 1 jof-09-00440-t001:** The Western Schley pecan orchards [*Carya illinoinensis* (Wangeh) K. Koch] with different agronomic management methods and ages in Chihuahua, Mexico, included in this study.

Orchard Name	Municipality	Latitude N/Longitude W Coordinates	Tree Age (Years)	Management Type	Soil Texture
El General	Saucillo	28°14′48″ 105°30′17″	3	Conventional	Sandy clay
El Escondido CM	Saucillo	28°04′42″ 105°19′53″	8	Conventional	Sandy clay
El Escondido SM	Saucillo	28°05′47″ 105°18′61″	8	Conventional	Sandy clay
San Jorge	Delicias	28°13′46″ 105°25′8″	10	Conventional	Sandy clay
La Reyna	Meoqui	28°14′47″ 105°30′20″	30	Conventional	Sandy crumb
La Carpintería	San Francisco de Conchos	27°33′4″ 105°24′18″	38	Conventional	Sandy crumb
Parcela Escolar	Aldama	28°45′50″ 105°57′54″	40	Conventional	Loam
4H	Saucillo	28°02′24″ 105°16′39″	6	Organic	Sandy crumb
El Maguey	Delicias	28°04′48″ 105°31′20″	9	Organic	Clayey crumb
Gaby	Meoqui	28°15′51″ 105°28′17″	40	Organic	Sandy crumb
Don Miguel	Saucillo	28°06′53″ 105°20′46″	47	Organic	Sandy crumb
Carmen MA	Jiménez	27°18′28″ 104°50′20″	48	Organic	Sandy clay crumb
Carmen SG	Jiménez	27°17′55″ 10°50′9″	48	Organic	Sandy clay crumb
La Concha	Chihuahua	9°03′26″ 106°11′55″	17	Organic	Sandy clay loam

**Table 2 jof-09-00440-t002:** Rhizospheric soils in 14 pecan orchards grouped by texture. Chihuahua, Mexico 2020.

Texture	% Ectomycorrhization	n = Number of Orchards
Clay–sandy	47.44	n = 1
Loamy sand	49.00	n = 3
Loam	31.44	n = 1
Loam–clay–sand	49.56	n = 1
Sandy crumb	48.65	n = 5
Sandy clay crumb	55.17	n = 2
Clayey crumb	34.11	n = 1

## Data Availability

Not applicable.

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
