# Peer review of "Soil Structure and Ectomycorrhizal Root Colonization of Pecan Orchards in Northern Mexico"

_jof, 2023, doi:10.3390/jof9040440_

Round 1
Reviewer 1 Report
Dear authors,
As a reviewer, I have carefully evaluated your work and provided feedback to the editor. While I appreciate the time and effort you've put into your writing, I regret to inform you that your manuscript needs improvement.
Kindly refer to the below feedback, and I encourage you to revise your manuscript. I hope that you find my feedback helpful in improving your work.
1. Abstract - The 1st sentence of your abstract is hard to understand. You may want to rephrase to something like this "Pecan trees form a symbiotic relationship with ectomycorrhizal fungi (ECMF) that actively provide nutrition to the roots and protect them from phytopathogens." Several other sentences in the abstract are also hard to understand. I would suggest you hire a proofreader for this matter.
2. The commonly used short form for ectomycorrhizal fungi is ECM, not EC. Please change.
3. Introduction - The introduction is weak and needs to be strengthened. More details are needed to establish a strong foundation for the article. In the fourth paragraph, you should have introduced ECM colonization. Only then could you mention changes in colonization that are due to several factors...
4. Introduction - Line 42 - 45. points need rearrangement
5. Materials and methodology - Lines 78-79 those methods must be cited accordingly.
6. Materials and methodology - Line 93 which August year you are referring to..
7. Results and Discussion - Line 138 - 139. Although you have provided the range of colonisation activity, it is recommended to display the percentage of root tips colonised by ECM for all 14 orchards.
8. Results and Discussion - subheading in line 178, 3.3.2, in my opinion, the subheading needs to be rephrased to conform to proper scientific context
Author Response
Response to Reviewer 1 Comments
Point 1: English language and style required
Response 1:
Thank you, we sent this final version to: Cambridge Proofreading & Editing.
US: 312-724-5771. UK & International: (+44) 03308 220012
Point 2. The 1st sentence of your abstract is hard to understand. You may want to rephrase to something like this "Pecan trees form a symbiotic relationship with ectomycorrhizal fungi (ECMF) that actively provide nutrition to the roots and protect them from phytopathogens." Several other sentences in the abstract are also hard to understand. I would suggest you hire a proofreader for this matter.
Response 2:
We replaced the old sentence for this: Pecan trees form a symbiotic relationship with ectomycorrhizal fungi (ECM) which actively provide nutrition to the roots and protect them from phytopathogens. Besides, we improved the whole abstract.
Point 3. for ectomycorrhizal fungi is ECM, not EC
Response 3: Done, thank you.
Point 4. The introduction is weak and needs to be strengthened. More details are needed to establish a strong foundation for the article. In the fourth paragraph, you should have introduced ECM colonization. Only then could you mention changes in colonization that are due to several factors...
Response 4:
We improved the introduction, added more sentences and now it is more strengthened. We added a paragraph about “ECM” before “changes in colonization”
Point 5. Introduction - Line 42 - 45. points need rearrangement
Response 5: We improved these lines in the introduction.
These are the new lines: ECM and the plant family Juglandaceae, including the genus Carya, often form sym-biotic associations [18]. In addition, the heat-tolerant and drought-tolerant genus Pisolithus forms a symbiosis with pecan tree roots [19-23].
Point 6. Materials and methodology - Lines 78-79 those methods must be cited accordingly.
Response 6: We added the references of Olsen & Sommers [38], and Bouyoucos [37]
Point 7: Materials and methodology - Line 93 which August year you are referring to.
Response 7: After “August” we already included “2020”
Point 8: Results and Discussion - Line 138 - 139. Although you have provided the range of colonisation activity, it is recommended to display the percentage of root tips colonised by ECM for all 14 orchards.
Response 8: We already indicated the % of the 14 orchards in Figure 2.
Point 9: Results and Discussion - subheading in line 178, 3.3.2, in my opinion, the subheading needs to be rephrased to conform to proper scientific context.
Response 9:
The subheading was: Organic matter (OM)-Phosphorus (P)-ectomycorrhization. The new one is: Influence of organic matter (OM) and Phosphorus (P) on ECM.
We substantially improved the entire manuscript, but if further changes are necessary, we are happy to do so. We consider it worthwhile to publish these results

Reviewer 2 Report
Dear Authors,
The subject you have selected is really interesting, but the paper must be improved. I think, the paper is worth to be published, but definitely not in its current form.
You made two major flaws. First is the lack of molecular identification of ECM fungi on the roots of pecan trees. Without molecular identification of ectomycorrhizas you provide no data on ECM associations, only root morphology and the accidental presence of two fungal sporocarps.
Can you improve it and provide molecular identification of ectomycorrhizas on pecan roots in tested orchards?
The second flaw is the methodology of sporocarp surveying - proper surveying requires numerous visits in autumn. You studied sporocarps in spring and summer when almost all ECM fungal species cannot produce sporocarps. I deduce you visited each orchard only one time. This is definitely not enough. As a result, you noted not several dozens or hundreds of ECM fungal sporocarps, but only two. And both belong to the heat-tolerant and drought-tolerant genus Pisolithus. Please, read chapter "Pisolithus" (Chambers & Cairney) in the book Ectomycorrhizal Fungi Key Genera in Profile [here: https://link.springer.com/chapter/10.1007/978-3-662-06827-4_1 ].
Please, describe precisely the month when the sporocarps were observed. Between March and August (line 58) is not precise. Moreover, autumn is the optimal season for ECM sporocarp surveying, but spring and summer are not. There is probably the answer to why you noted only 2 sporocarps.
Further comments and objections are below.
Title
This title is not appropriate. First, you did not study ECM associations because no molecular identification of ectomycorrhizas was provided in this study.
The proper title of research articles should involve the object of the study and the location (if it is not a global-scale meta-analysis). So I propose a title: Soil structure and ectomycorrhizal root colonization in pecan orchards in northern Mexico
INTRODUCTION
Line 19
Please, describe how much pecan nuts are produced in the USA versus Mexico. Here is given "the U.S. production estimate was about 75% of the world's pecans and Mexico about 20%". Is it true? https://www.farmprogress.com/orchard-crops/u-s-loses-top-pecan-production-spot-to-mexico
Lines 42-45
I think, that all available data on ECM fungi associated with hickory trees should be referenced here. There are not many papers about ECM associations formed by any hickory species, so all published papers should be referenced. Studies in Europe have shown, that hickory trees (Carya ssp., native to North America, alien in Europe) can easily establish ECM symbiosis with local ECM fungi (Rudawska et al. 2018, Wilgan et al. 2020, Wilgan, Leski 2022). So if hickory trees can establish symbiosis with numerous local ECM fungi in Europe, there is almost certain, there will find a rich pool of appropriate ECM fungal symbionts in their native range.
Moreover, European fungal species from Sclerodermataceae were almost absent on the roots of hickories in Europe, although they were abundant on Q. rubra in the direct vicinity (Wilgan, Leski 2022). This result would be interesting for the discussion of your results for Pisolithus (look at the discussion in Wilgan, Leski 2022).
Rudawska, M., Leski, T., Wilgan, R. et al. 2018. Mycorrhizal associations of the exotic hickory trees, Carya laciniosa and Carya cordiformis, grown in Kórnik Arboretum in Poland. Mycorrhiza 28, 549–560. https://doi.org/10.1007/s00572-018-0846-8
Wilgan, R.; Leski, T.; Kujawska, M.; Karliński, L.; Janowski, D.; Rudawska, M. 2020. Ectomycorrhizal fungi of exotic Carya ovata in the context of surrounding native forests on Central European sites. Fungal Ecol. 44, 100908. https://doi.org/10.1016/j.funeco.2019.100908
Wilgan R., Leski T. 2022, Ectomycorrhizal Assemblages of Invasive Quercus rubra L. and Non-Invasive Carya Nutt. Trees under Common Garden Conditions in Europe. Forests 13(5), 676.
Line 56
I suggest modifying line 56 as follows:
The study included 14 pecan orchards in Chihuahua state in Mexico.
The Latin name for pecan tree was already used, there is no requirement to use it again, and in this form, the sentence is more clear.
Line 67: Table 1
Please, rewrite this table. The current form (scan) is poorly readable and moreover, it does not fulfill the MDPI requirements for the accepted resolution of figures. (This is a figure - scan of a table.)
Moreover please, add a map with these locations. How many 8x8 and 13x13 frames were studied in the individual pecan orchard?
Line 69
<The owners of Carmen MA, Carmen SG and San Jorge orchards previously applied commercial ectomycorrhizal fungi.>
Which fungi? Did you use truffle orchards with pecan trees? Please, describe it briefly.
RESULTS AND DISCUSSION
I recommend the Authors separate the results and discussion.
First, you refer to studies on alder (Alnus), lime (Tilia), and spruce (Picea). That's wrong. First, Alnus for ECM and AM associations, and moreover, symbiosis with nitrogen-fixing bacteria, while Picea and Tilia belong to distinct tree orders (Malvales and Pinales, respectively). I suggest you refer to studies on Carya trees or other Fagales genera (e.g. Quercus, Fagus, Carpinus, Castanea).
CONCULSIONS
Line 336
"the pecan tree in the south-central region of Chihuahua, Mexico is susceptible to colonization by ECM fungi" - this sentence is clumsy.
"Susceptible" is not a proper word. Hickories require ECM fungi to survive and develop. ECM symbiosis is mandatory for hickories, same as for oaks, beech, chestnut, and the majority of ECM tree genera.
This is the most important result the presented paper provides:
"The variability of the EC percentage can be influenced by several factors, highlighting the phosphorus content, which is negatively related to the ECM colonization percentage. Similarly, adult trees with organic management have a positive correlation with the percentage of ECM. Moreover, in young trees, agronomic management did not show a significant statistical effect with respect to the percentage of colonization by ECMF."
Line 347
"These results will allow the interrelation of pecan trees genetics and its microbiome because this symbiosis creates a very specific microenvironment. "
Did you study pecan genetics? This sentence does not belong to the conclusion. If anything, dry and warm environmental conditions (and mismatched season of sporocarp surveying!) generate microniches for specific ECM fungi, such as drought-tolerant and heat-tolerant Pisolithus species.
Author Response
Response to Reviewer 2 Comments
Point 1:
You made two major flaws. First is the lack of molecular identification of ECM fungi on the roots of pecan trees. Without molecular identification of ectomycorrhizas you provide no data on ECM associations, only root morphology and the accidental presence of two fungal sporocarps.
Can you improve it and provide molecular identification of ecotomycorrhizas on pecan roots in tested orchards?
Response 1: The reviewer is right, more studies need to be carried out on the identification of ECM using molecular genetic analysis in the root of pecan trees. Since nowadays the information is scarce, our research group is planning this identification in the next project.
However, we believe that our scientific contribution that we report in this study is very valuable, since only 4 Pisolithus sp ITS gene sequences are deposited in the GenBank, three from Chihuahua, Mexico and one from Brazil (associated with Carya). Two of the three Mexican sequences are the ones reported in this study.
Point 2. The second flaw is the methodology of sporocarp surveying - proper surveying requires numerous visits in autumn. You studied sporocarps in spring and summer when almost all ECM fungal species cannot produce sporocarps. I deduce you visited each orchard only one time. This is definitely not enough. As a result, you noted not several dozens or hundreds of ECM fungal sporocarps, but only two. And both belong to the heat-tolerant and drought-tolerant genus Pisolithus. Please, read chapter "Pisolithus" (Chambers & Cairney) in the book Ectomycorrhizal Fungi Key Genera in Profile [here: https://link.springer.com/chapter/10.1007/978-3-662-06827-4_1
Response 2:
Thank you for providing the link of this chapter, it is very useful literature. We included it in our manuscript. We colected the two typical sporocarps specimens of Pisolithus, the fungus previously identified as Carya's ectomycorrhizal. Future studies will focus, first of all, on the metagenomics of ectomycorrhizal roots.
Point 3. Please, describe precisely the month when the sporocarps were observed. Between March and August (line 58) is not precise. Moreover, autumn is the optimal season for ECM sporocarp surveying, but spring and summer are not. There is probably the answer to why you noted only 2 sporocarps.
Response 3:
At the end of August the sporocarps were collected. Between March and August we collected the root tips in the 14 orchards. We now clarified this in lines 71 and 121.
Point 4:
Title
This title is not appropriate. First, you did not study ECM associations because no molecular identification of ectomycorrhizas was provided in this study.
The proper title of research articles should involve the object of the study and the location (if it is not a global-scale meta-analysis). So I propose a title: Soil structure and ectomycorrhizal root colonization in pecan orchards in northern Mexico.
Response 4: Thank you. The authors agree that the title you propose is convenience.
Point 5.
INTRODUCTION
Line 19. Please, describe how much pecan nuts are produced in the USA versus Mexico. Here is given "the U.S. production estimate was about 75% of the world's pecans and Mexico about 20%". Is it true? https://www.farmprogress.com/orchard-crops/u-s-loses-top-pecan-production-spot-to-mexico.
Response 5: We already incorporated production in Mexico and USA in 2017. USA import large amounts of pecans and then export. Moreover, Mexico has the largest production worldwide.
Point 6.
Lines 42-45
I think, that all available data on ECM fungi associated with hickory trees should be referenced here. There are not many papers about ECM associations formed by any hickory species, so all published papers should be referenced. Studies in Europe have shown, that hickory trees (Carya ssp., native to North America, alien in Europe) can easily establish ECM symbiosis with local ECM fungi (Rudawska et al. 2018, Wilgan et al. 2020, Wilgan, Leski 2022). So if hickory trees can establish symbiosis with numerous local ECM fungi in Europe, there is almost certain, there will find a rich pool of appropriate ECM fungal symbionts in their native range.
Moreover, European fungal species from Sclerodermataceae were almost absent on the roots of hickories in Europe, although they were abundant on Q. rubra in the direct vicinity (Wilgan, Leski 2022). This result would be interesting for the discussion of your results for Pisolithus (look at the discussion in Wilgan, Leski 2022).
Rudawska, M., Leski, T., Wilgan, R. et al. 2018. Mycorrhizal associations of the exotic hickory trees, Carya laciniosa and Carya cordiformis, grown in Kórnik Arboretum in Poland. Mycorrhiza 28, 549–560. https://doi.org/10.1007/s00572-018-0846-8
Wilgan, R.; Leski, T.; Kujawska, M.; Karliński, L.; Janowski, D.; Rudawska, M. 2020. Ectomycorrhizal fungi of exotic Carya ovata in the context of surrounding native forests on Central European sites. Fungal Ecol. 44, 100908. https://doi.org/10.1016/j.funeco.2019.100908
Wilgan R., Leski T. 2022, Ectomycorrhizal Assemblages of Invasive Quercus rubra L. and Non-Invasive Carya Nutt. Trees under Common Garden Conditions in Europe. Forests 13(5), 676
Response 6: Yes you are right.
These references are value and we already included in this study.
In the study of Wilgan and Leski (2022) on Carya, the authors mentioned that the ECM are abundant up to 25 taxa in plants, and 40 taxa in mature trees, however, they do not report Pisolithus in Carya.
Instead we compared by multiple sequence analysis the 4 sequences of Pisolithus in Carya in the GenBank (NCBI) including the Brazilian strain MT586545.1
In relation to the fungus-root association, a future metagenomic analysis of ectomycorrhizal roots will definitely bring light and generate knowledge about this symbiosis.
Point 7.
Line 56. I suggest modifying line 56 as follows:
The study included 14 pecan orchards in Chihuahua state in Mexico.
The Latin name for pecan tree was already used, there is no requirement to use it again, and in this form, the sentence is more clear.
Response 7:
Thank you, now we have modified the sentence. We deleted the latin name in this line.
Point 8.
Line 67: Table 1
Please, rewrite this table. The current form (scan) is poorly readable and moreover, it does not fulfill the MDPI requirements for the accepted resolution of figures. (This is a figure - scan of a table.)
Moreover please, add a map with these locations. How many 8x8 and 13x13 frames were studied in the individual pecan orchard?
Response 8:
Done, new Table is not an image. Regardless of the frame, 3 trees were studied per orchard. We added the Figure 1 to illustrate the frames. Is it correct ?
Point 9.
Line 69
<The owners of Carmen MA, Carmen SG and San Jorge orchards previously applied commercial ectomycorrhizal fungi.>
Which fungi? Did you use truffle orchards with pecan trees? Please, describe it briefly.
Response 9:
The commercial product added in these orchards contained: Pisolithus tinctotium 1x106 spores/g, Glomus intradarises 1x103 spores/g , Azospirillum brasilense 1x106 CFU/g , total oxidizable organic carbon 20%
Point 10.
RESULTS AND DISCUSSION
I recommend the Authors separate the results and discussion.
First, you refer to studies on alder (Alnus), lime (Tilia), and spruce (Picea). That's wrong. First, Alnus for ECM and AM associations, and moreover, symbiosis with nitrogen-fixing bacteria, while Picea and Tilia belong to distinct tree orders (Malvales and Pinales, respectively). I suggest you refer to studies on Carya trees or other Fagales genera (e.g. Quercus, Fagus, Carpinus, Castanea).
Response 10:
Thank you, we already separated Results and Discussion, the manuscript now is more clear. We eliminated the studies of Alnus, Tilia and Picea. Instead we focussed on Carya.
Point 11.
CONCLUSIONS
Line 336
"the pecan tree in the south-central region of Chihuahua, Mexico is susceptible to colonization by ECM fungi" - this sentence is clumsy.
"Susceptible" is not a proper word. Hickories require ECM fungi to survive and develop. ECM symbiosis is mandatory for hickories, same as for oaks, beech, chestnut, and the majority of ECM tree genera.
This is the most important result the presented paper provides:
"The variability of the EC percentage can be influenced by several factors, highlighting the phosphorus content, which is negatively related to the ECM colonization percentage. Similarly, adult trees with organic management have a positive correlation with the percentage of ECM. Moreover, in young trees, agronomic management did not show a significant statistical effect with respect to the percentage of colonization by ECMF."
Response 11:
Thank you for these observations. The new conclusions are improved.
We eliminated the word “susceptible”
Yes, we think our study is valuable since here is reported that the percentage of ECM is influenced by factors such as the phosphorus content, which is negatively related to the percentage of ECM colonization. Besides, that adult trees with organic management have a positive correlation with the percentage of ECM, but not young trees.
Point 12:
Line 347
"These results will allow the interrelation of pecan trees genetics and its microbiome because this symbiosis creates a very specific microenvironment. ".
Did you study pecan genetics? This sentence does not belong to the conclusion. If anything, dry and warm environmental conditions (and mismatched season of sporocarp surveying!) generate microniches for specific ECM fungi, such as drought-tolerant and heat-tolerant Pisolithus species.
Response 12:
You are right, we deleted this part.
We substantially improved the entire manuscript, but if further changes are necessary, we are happy to do so. we consider it worthwhile to publish these results.

Round 2
Reviewer 2 Report
The Authors have referred to all my comments. I am satisfied with the response to the review.
I think particular parts of the paper can still be improved. However, I see no other crucial contraindications nor substantial methodical errors than previously described, so I marked "accept in present form".
I hope the next study on ECM roots of Carya trees with a metagenomic approach will answer further questions.
The comparative studies between wild Carya stands and Carya plantations under forest/agricultural management would be interesting.
I think this paper on the relic tree family Juglandaceae and its conservation priorities would be inspiring to the Authors:
https://onlinelibrary.wiley.com/doi/full/10.1111/jbi.13766
This is a good paper, but unfortunately, the mycorrhizal symbiosis of Juglandaceae trees was entirely omitted by Song et al. (2020).